# Towards an Approach for Filtration Efficiency Estimation of Consumer-Grade Face Masks Using Thermography

**José Armando Fragoso-Mandujano** [1], **Madain Pérez-Patricio** [1,*], **Jorge Luis Camas-Anzueto** [1], **Hector Daniel Vázquez-Delgado** [1,*], **Eduardo Chandomí-Castellanos** [1], **Yair Gonzalez-Baldizón** [1], **Julio Alberto Guzman-Rabasa** [1], **Julio Cesar Martinez-Morgan** [1] and **Luis Enrique Guillén-Ruíz** [1]

Tecnológico Nacional de México, I.T Tuxtla Gutiérrez, Posgrado en Ciencias de la Ingeniería, Carretera Panamericana KM. 1080, Tuxtla Gutiérrez 29050, Mexico; d07270448@tuxtla.tecnm.mx (J.A.F.-M.); jcamas@tuxtla.tecnm.mx (J.L.C.-A.); eduardo.cc@tuxtla.tecnm.mx (E.C.-C.); ygbaldizon@cenidet.edu.mx (Y.G.-B.); jaguzmanrabasa@gmail.com (J.A.G.-R.); cesarmtzmorgan@gmail.com (J.C.M.-M.); lmcguillen@live.com.mx (L.E.G.-R.)
* Correspondence: madain.pp@tuxtla.tecnm.mx (M.P.-P.); hector.vazquez@ittg.edu.mx (H.D.V.-D.)

**Abstract:** Due to the increasing need for continuous use of face masks caused by COVID-19, it is essential to evaluate the filtration quality that each face mask provides. In this research, an estimation method based on thermal image processing was developed; the main objective was to evaluate the effectiveness of different face masks while being used during breathing. For the acquisition of heat distribution images, a thermographic imaging system was built; moreover, a deep learning model detected the leakage percentage of each face mask with a mAP of 0.9345, recall of 0.842 and F1-score of 0.82. The results obtained from this research revealed that the filtration effectiveness depended on heat loss through the manufacturing material; the proposed estimation method is simple, fast, and can be replicated and operated by people who are not experts in the computer field.

**Keywords:** COVID-19; thermography; face mask; filtration efficiency

## 1. Introduction

Coronavirus disease 2019 (COVID-19) is a highly contagious and pathogenic viral infection caused by severe acute respiratory syndrome coronavirus 2 (SARS-CoV-2), first reported in Wuhan, China, and currently widespread around the world [1,2]. COVID-19 has been spread primarily from person to person through tiny droplets of fluid expelled from the nose or mouth of an infected person by coughing, sneezing, or talking [3].

According to the World Health Organization (WHO), by ensuring strict hygiene measures, correct handwashing, and a distance between people of at least 1.5 m, the transmission of the virus is reduced [4,5]. Furthermore, governments around the world have recommended the use of face masks to all their citizens [6], because face masks prevent the infection of COVID-19 between people [7]. However, face masks' effectiveness in preventing the spread of COVID-19 and other respiratory diseases has decreased mainly due to their misuse and poor fitting [8], including low-quality manufacturing processes and unsuitable materials [7].

The pandemic caused by COVID-19 has opened a path to the developing technologies in the field, such as deep learning and artificial intelligence (AI), both of these have made everyday life easier by providing solutions to several complex problems in different areas [9]. Modern computer vision algorithms are approaching human-level performance in visual perception tasks.

Convolutional neural networks (CNN) are essential in artificial vision processes to detect objects, monitor, and classify images, among others. A CNN uses convolution kernels to extract higher-level features from original images or feature maps, resulting in a powerful tool for computer vision tasks. Computer vision has proven to be a revolutionary

aspect of modern technology in a world battling the pandemic [10]. Furthermore, deep learning has allowed researchers and clinicians to evaluate large amounts of data to forecast the propagation of COVID-19, running as an early warning mechanism for potential pandemics and classifying vulnerable populations. AI addresses and predicts new diseases by understanding infection rates, helping to provide a fully automatic and quick diagnosis for COVID-19 from X-ray images [11], and evaluates the prediction performance of death by COVID-19 based on the demographic and clinical factors [12], among others [13]. In this sense, to address the COVID-19 pandemic, AI's intrinsic benefits are being harnessed [14].

The current development of artificial intelligence has allowed researchers to provide an approach for automatic detection of the conditions of the appropriate use of face masks and distancing [15], which in combination with the super-resolution of images with a classification network (SRCNet), can contribute through technological innovations in deep transfer learning and computer vision [8] to personal protection against and public prevention of epidemics. In [14], the authors presented a model that used classic and deep machine learning to detect face masks. The proposed research consisted of two parts. The first component focused on feature extraction using Resnet50; the second component classified face masks using decision trees, support vector machine (SVM), and ensemble algorithm. Furthermore, in [16], the authors used the YOLOV v3 algorithm with some improvements, adjusting the algorithm to detect small faces. The proposed method was trained on two databases: WIDER FACE and CelebA, and it was also tested on the FDDB database, achieving a precision of 93.9%. Furthermore, recently, there has been some research about the effectiveness of the proper use of face masks and their variations. Although previous studies have shown the effectiveness of using face masks to filter particles, each face mask's effectiveness depends on its characteristics [17].

A particle generator with a mean diameter of 0.05 μm was used to measure face masks' filtering effectiveness in [18]; the authors equipped each face mask with sampling probes to take particle samples within the face mask, and used condensation particle counters to monitor particles (0.02–3.00 μm) in the environment and behind the face mask.

Although the studies previously mentioned supported face masks' potential beneficial effect, the substantial impact of face masks on the spread of laboratory-diagnosed respiratory viruses has remained controversial due to their characteristics in terms of materials and quality [3], thus requiring new analyses with different approaches, making use of deep learning and image processing.

The upper airway heats and humidifies the air reaching the lungs at the internal body temperature (37 °C) during inhalation. When exhaling, the mucosa only recovers part of the heat and humidity added during inhalation, which causes a significant loss of heat and moisture to the environment; therefore, both the loss of heat and moisture changes can demonstrate the ineffectiveness of a face mask [19].

This research presents a computer vision system based on infrared thermography and deep learning to estimate the filtration quality of different consumer face masks. This system allows images of the air distribution that pass through the face masks; once the images are obtained, they are compared with the training images selected for the CNN to know which face masks provide the best protection against the dispersion of SARS-CoVo2 in the air.

The main research contributions are as follows:

- A novel method for the estimation, evaluation, and custom classification of face masks.
- The image acquisition and processing system are fast, low-cost, versatile, and scalable to other platforms and thermal sensors.
- A qualitative demonstration of filtration efficiency is presented.

The rest of the article organize as follows:

In Section 2, the materials and methods used in this work are presented, and the experimental process is described. Section 3 describes the results and discuss them. Finally, Section 5 presents the conclusions obtained in this article.

## 2. Materials and Methods

### 2.1. Thermographic Imaging System

For the heat distribution images acquisition, a test bench was built following the scheme shown in Figure 1; an embedded system (Raspberry Pi 3 Model B +, Raspberry Pi Foundation) was used with Python 3 installed as the development language. Images were acquired using a shutterless radiometric thermal module (FLIR Lepton 2.5, FLIR Systems, Wilsonville, OR, USA). The FLIR Lepton module had a resolution of $80 \times 60$ pixels resized via software (Open CV library for Python) to $618 \times 450$ pixels, an 8–14 µm spectral range, and thermal sensitivity of 0.050 °C. A heat diffusion screen made of a wooden frame covered with kraft paper 125 g/m$^2$) was made to make the residual air heat distribution visible, and it was placed at a distance of 10 cm from the test subject. The thermal properties of the kraft paper are presented in Table 1; finally, the thermal camera was placed 25 cm away in front of the frame.

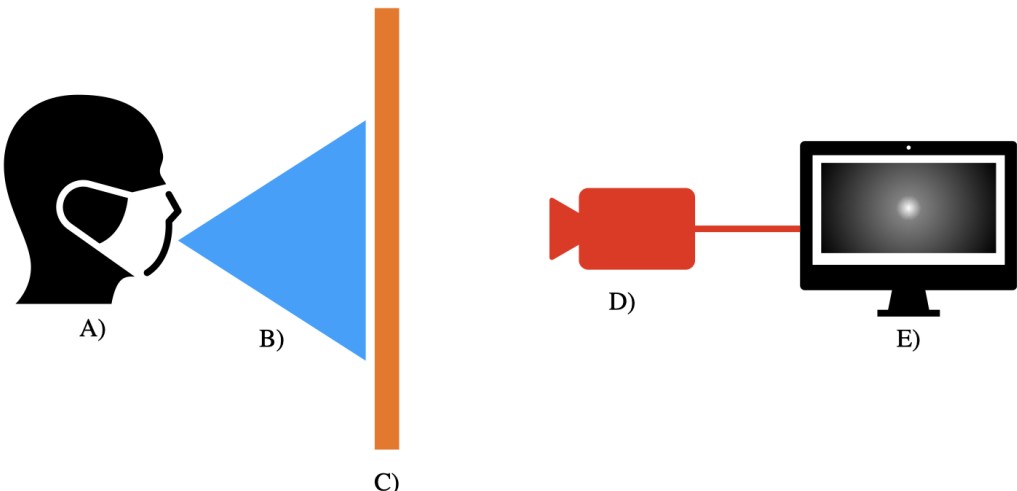

**Figure 1.** General configuration of the experiments: (**A**) test subject and heat source; (**B**) hot air; (**C**) heat diffusion screen; (**D**) thermal camera module; (**E**) embedded system to display data, record, and process.

**Table 1.** Thermal properties of the Kraft-paper ™ [20].

| Thermal Property | |
|---|---|
| $k$ (W/m K) | 0.066 |
| $\rho$ (kg/m$^3$) | 104 |
| $Cp$ (J/kg K) | 1355 |
| $\alpha$ (m$^2$/s) | $4.70 \times 10^{-7}$ |

### 2.2. Testing Procedure

The leakage tests were carried out between August and November 2020 in an exposure chamber (Tecnológico Nacional de México/I.T de Tuxtla Gutiérrez. In Tuxtla Gutiérrez, Chiapas). The following environmental conditions were maintained to avoid reflections and temperature changes inside the exposure chamber: temperature and humidity during the tests ranged from 23 °C to 28 °C and 10% to 50%, respectively. The environmental test conditions used for this study were laboratory conditions. In this study, all the face masks were fitted on a man with an 85 kg weight, a 180 cm height, and a 60 cm head size.

The testing bench collected temperature distribution videos of the air that passed through each face mask; to obtain this, the test subject performed a series of repeated movements of the torso, head, and facial muscles as described in the Occupational Safety and Health Administration (OSHA) quantitative fit test protocol [21] to simulate the typical occupational activities experienced by a face mask user. The total testing time for each

face mask was approximately three minutes. The measurements were made every 20 min, sufficient for the diffusion screen to reach room temperature.

### 2.3. Products Tested

One characteristic of consumer-grade face masks is the lack of quantitative information on filtration efficiency. In general, the face masks specifications do not have information about the porosity of the fabric and the particle size allowed to pass through the fabric. We tested the following consumer face masks for this study (Figure 2): (1) a two-layer woven cotton face mask (98% cotton, 2% spandex) with ear loops, tested without an aluminum nose bridge (Figure 2a), (2) a two-layer woven polyester face mask (100% polyester) (Figure 2b) with ear loops and tested without a nose bridge, (3) a three-layer nonwoven face mask made of neoprene (100% neoprene) with fixed ear loops and tested without a nose bridge (Figure 2c), (4) a three-layer nonwoven face mask made of melt-blown fabric (100% nonwoven melt-blown fabric) with elastics ear loops and aluminum nose bridge (Figure 2d), (5) a three-layer face mask of woven cotton (100% cotton) with fixed ear loops and plastic nose bridge (Figure 2e), and (6) a three-layer surgical face mask (70% nonwoven, 30% melt-blown fabric) with elastics ear loops and aluminum nose bridge (Figure 2f).

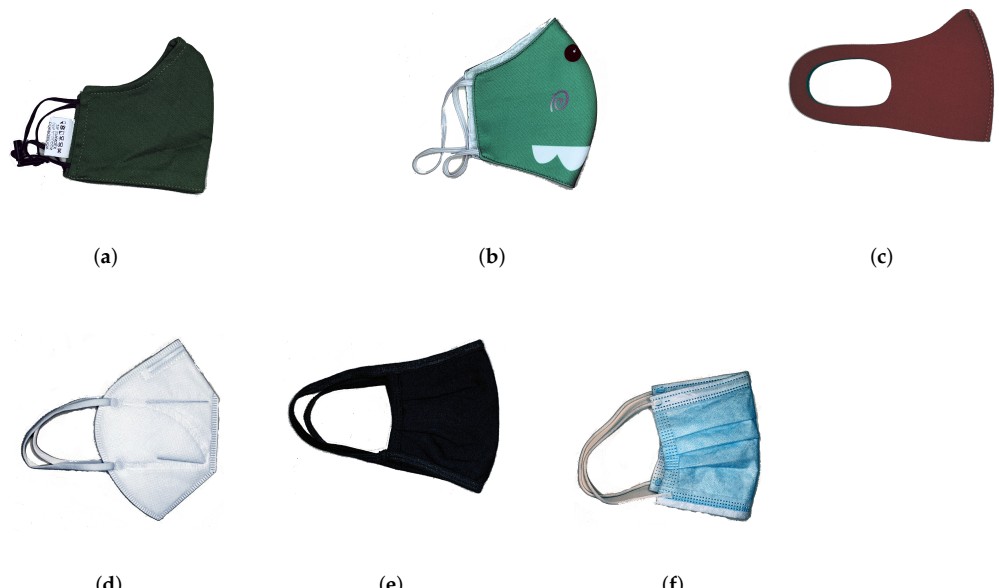

(a)      (b)      (c)

(d)      (e)      (f)

**Figure 2.** Consumer-grade face masks. (**a**) 98% cotton; (**b**) 100% polyester; (**c**) neoprene; (**d**) KN95; (**e**) 100% cotton; (**f**) 3-layer pleated.

### 2.4. Leakage Percentage

The thermal module translates the environmental temperature information into a gray intensity matrix, where the ambient temperature is zero and the maximum temperature recorded in the scene is 255; therefore, for the approximation presented in this work, some thermal parameters were neglected during the measures. From each test, a video was produced; a three-dimensional matrix $T(x, y, z) \in \mathbb{R}^{mxnxp}$ was obtained from each sequence of images. All pixels in each image were averaged together according to Equation (1). The result was an average value for each image; each value was stored in a .CSV file and plotted versus time to obtain the temperature's temporal behavior (Figure 3).

$$\bar{a}_k = \frac{1}{n \cdot m} \sum_{i=1}^{m} \sum_{j=1}^{n} T_{i,j}, \text{ for all } k \in \{1, 2, 3, 4, \ldots, p\}, \tag{1}$$

where $i$ and $j$ represents the pixel coordinates, $m$ and $n$ the image dimensions, and $\bar{a}_k$ is the gray intensity values summary of each image pixel divided by the total number of pixels $(n \cdot m)$ for each video frame ($k$).

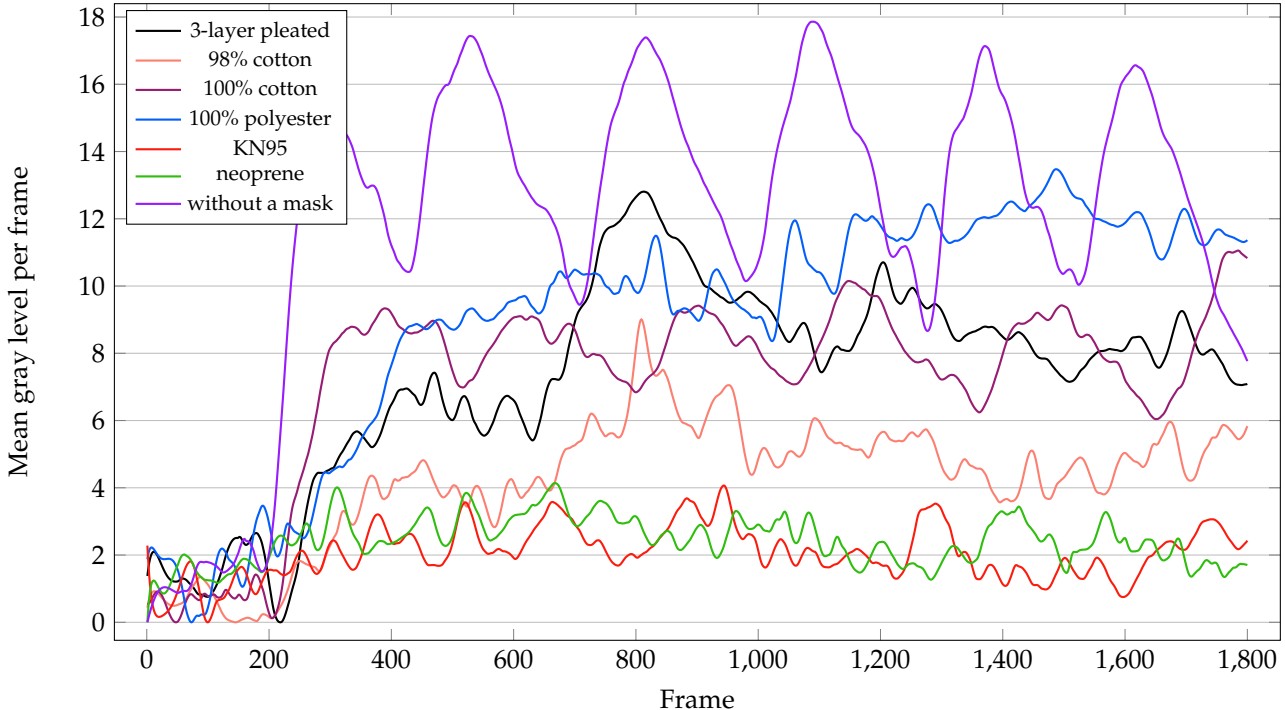

**Figure 3.** Temperature's temporal behavior by product.

The obtained data were rearranged in such a way that all the tests done without wearing a face mask were considered as a 100% leak; taking into consideration its mean value of gray and its corresponding leakage percentage, we were able to estimate the leakage percentages of each face mask. The summary of the data is shown in Table 2. The values were taken from the leakage percentage. column to label the training images and the test images.

**Table 2.** Summary of information for each face mask.

| Face Mask | Mean Pixels | Leakeage Percentage |
|---|---|---|
| Without a mask | 46.6 | 100% |
| 100% cotton | 32.43 | 69% |
| 100% polyester | 28.37 | 60% |
| 3-layer pleated | 18.28 | 39% |
| 98% cotton | 12.49 | 26% |
| KN95 | 11.99 | 25% |
| Neoprene | 22.55 | 25% |
| Mask without any leak | 0 | 0% |

### 2.5. Data Set

The thermal images used extracts from previously recorded videos. The whole of this study had a total of 172 images with 618 pixels (horizontal) $\times$ 450 pixels (vertical) resolution, 138 of which (80%) were labeled as training images, and 34 (20%) remained as test data. We used *LabelImg* to manually label these 172 images, ensuring that each image's temperature focus was located in the labeling box's center. However, to improve detection, we opted for the generation of artificial information.

The basic generation of artificial information was carried out as follows:

Reflection X: each image was flipped vertically (Equation (2)); reflection Y: each image was flipped horizontally (Equation (3)); reflection XY: each image was flipped vertically and horizontally (Equation (4)).

$$\begin{bmatrix} x' \\ y' \end{bmatrix} = \begin{bmatrix} 1 & 0 \\ 0 & -1 \end{bmatrix} \begin{bmatrix} x \\ y \end{bmatrix} \tag{2}$$

$$\begin{bmatrix} x' \\ y' \end{bmatrix} = \begin{bmatrix} -1 & 0 \\ 0 & 1 \end{bmatrix} \begin{bmatrix} x \\ y \end{bmatrix} \tag{3}$$

$$\begin{bmatrix} x' \\ y' \end{bmatrix} = \begin{bmatrix} -1 & 0 \\ 0 & -1 \end{bmatrix} \begin{bmatrix} x \\ y \end{bmatrix} \tag{4}$$

where $x$ and $y$ represents the horizontal and vertical axis, respectively, on the image.

Rotation: each image changed its axis. For this rotation, the rotation matrix given in Equation (5) was used, where $\theta$ represents the rotation angle of the image and $R(\theta)$ the new pixel position.

$$R(\theta) = \begin{bmatrix} \cos\theta & -\sin\theta \\ \sin\theta & \cos\theta \end{bmatrix} \tag{5}$$

Median filter: each pixel in the image is replaced with the median value of its neighboring pixels ($K$) in the image. The operation is represented by Equation (6).

$$K = \frac{1}{9} \begin{bmatrix} 1 & 1 & 1 \\ 1 & 1 & 1 \\ 1 & 1 & 1 \end{bmatrix} \tag{6}$$

Combining the fundamental data augmentation transformations (i.e., reflection x + median filter, rotation + median filter, among others) enlarged the data set, in our case, nine times larger than the original size. The additional training data helped the model avoid the overfitting that occurs when training with small amounts of data. Therefore, the data augmentation helped build a simple, robust, and generalizable model. Concerning the artificial data, 80.0% (1247 images) were used as training data for the object detection algorithm, and the remaining 20.0% (311 images) were used as test data to experiment and verify. Figure 4 shows the results of the labeling of two percentages of air filtration.

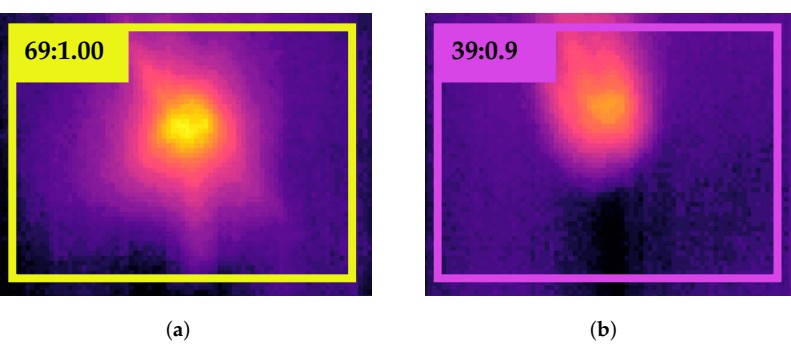

(a)          (b)

**Figure 4.** Examples of labeled images. (**a**) Label, 60% leak; (**b**) Label, 36% leak.

### 2.6. Deep Learning System

The term deep learning system refers to a feedforward neural network with a deep multilayer hierarchical structure [22].

Images are provided to the model through an input layer. Moreover, an output layer returns the detected object's category and its respective confidence score within bounding boxes. Between the input and output layers, there are hidden layers. Each layer consists of nodes connected to the nodes of the previous layers by weighted edges [23].

The YOLO (you only look once) network is a deep learning algorithm for detecting objects in one stage (Figure 5). It uses a single CNN to process images, and it can directly calculate the classification and position coordinates of objects. With the positioning and objects classification from one end to the other, the detection speed increases considerably [24]. Additionally, [25] confirms that YOLO v4 tends to be the best object detector in terms of accuracy and speed of object detection.

Figure 6 illustrates the proposed general deep learning model. Furthermore, to perform detection based on YOLO v4, the following steps were introduced:

Data organization: after collecting the data from the experimental images, all the temperature distribution images from the thermographic imaging system were manually labeled to complete the data set preparation.

Data augmentation: the data set was processed to generate artificial information, then the data set was divided into a training set (1247 images) and a test set (311 images).

In configuration and network training, the model parameters were adjusted using the pretrained YOLO v4 model, especially the batch size, the learning rate, the number of category objects, and the number of iterations.

The network's input size was 416 × 416 pixels (preset parameter by the YOLO v4). The YOLO v4 parameters in this study are shown in Table 3.

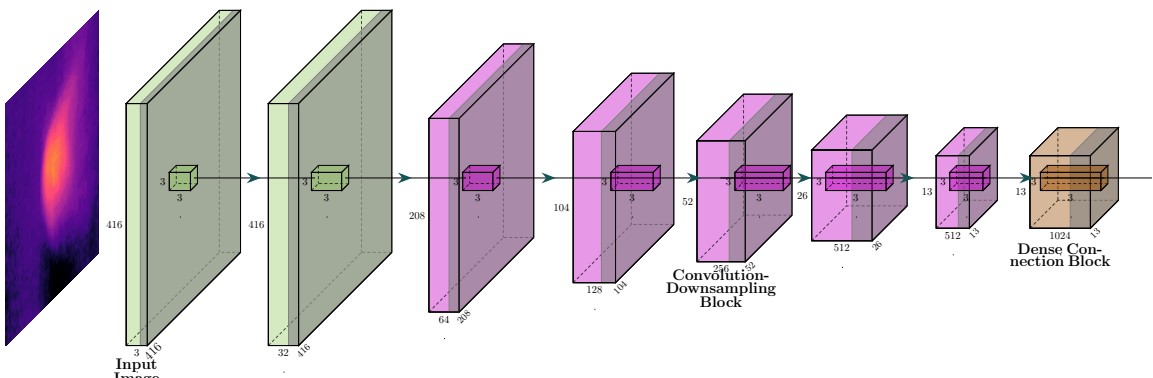

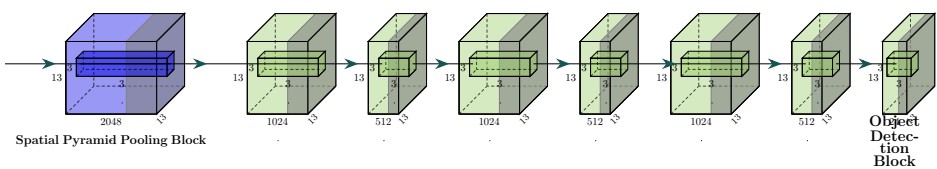

**Figure 5.** YOLO v4 general architecture.

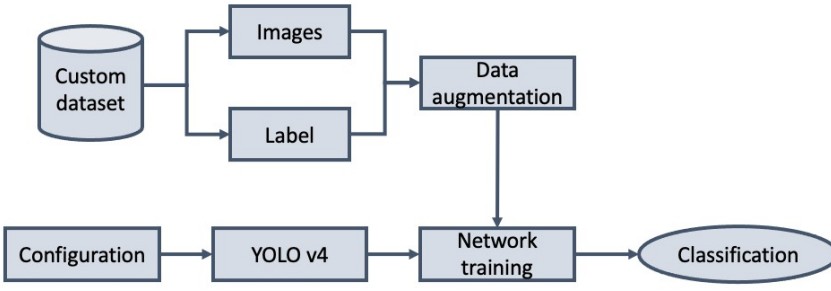

**Figure 6.** The proposed deep learning model.

For this research, we used Nvidia CUDA v10.1 on a Tesla T4 (MSI, Taiwan) graphics board provided by Google Colab servers.

**Table 3.** Parameters of the YOLO v4 face masks leakage model.

| Parameters | Value |
|---|---|
| Input size | $416 \times 416$ |
| Learning rate | $1 \times 10^{-3}$ |
| Batch size | 64 |
| Categories | 7 |
| Iterations | 5000 |

*2.7. Evaluation Procedure and Metrics*

In this research, the YOLO v4 deep learning system evaluated the model performance using the confusion matrix terminology [26] shown below:

- True positive ($TP$): it occurs when an object category is detected, and the image contains this object class in the indicated position.
- False positive ($FP$): it means that an object category is detected, but this object class is not in the position indicated in the image.
- False negative ($FN$): it occurs when an object category is in a specific position, and the model cannot detect it.
- True negative ($TN$): no object category is in that specific position, and the model did not detect any object.

The annotation and the bounding box's expected shape did not match perfectly during object detection, so an additional parameter was required to calculate the mentioned variables. This parameter was called intersection over union (IoU) and determined the required relative overlap $\alpha$ of the shape of the bounding boxes $B_p$ and the ground truth $B_g t$ as defined by [27] in Equation (7):

$$\alpha = \frac{area\left(B_p \cap B_{gt}\right)}{area\left(B_p \cup B_{gt}\right)} \tag{7}$$

The default value of this parameter is 0.5 [28]. Using the terminology of the confusion matrix and IoU , the following metrics were calculated [26]:

$$Recall = \frac{TP}{TP + FN} \tag{8}$$

$$Precision = \frac{TP}{TP + FP} \tag{9}$$

Average precision ($AP$) measures an object detector's performance related to a specific category in the object detection task. The procedure to calculate the $AP$ was as follows:

1. All the detections were sorted based on the confidence score.
2. The detections with the highest confidence score were matched to the ground truth until a recall $r$ higher than the expected $r$ level was reached.
3. Precision values based on each level of recall $r$ were calculated.
4. The precision $P_{interp}$ was interpolated by the maximum precision obtained for a recall level $r$.

The precision $P_{interp}$ is defined by [28] in Equation (10):

$$P_{interp}(r) = \max_{r:r \geq r} p(r) \tag{10}$$

where $p(r)$ is the measured precision at recall $r$.

For this research, we used eleven levels of recall $r \in \{0, 0.1; ..., 1\}$ with the same distance among them. Finally, we used $AP$, the arithmetic mean of the precision $P_{interp}$ at different levels of *Recall* [28] as shown in Equation (11):

$$AP = \frac{1}{11} \sum_{r \in \{0,0.1;...,1\}} P_{interp}(r). \tag{11}$$

Furthermore, the mean average precision (mAP) (Equation (12)) is the mean of the AP values for each object category [29], and the higher the value, the better the result of detecting temperature distributions.

$$mAP = \frac{\sum_{c=1}^{C} AP(c)}{C} \tag{12}$$

where $C$ is the number of detection categories. For the specific case of this study, $C = 7$.

### 3. Results

Figure 7 showed a representative image of the thermographic spectrum corresponding to each test. In tests without face mask restriction, 100% is considered (see Figure 7a). In descending order, the 100% cotton face mask presented a 69% leakage (see Figure 7b); the 100% polyester face mask showed 60% leakage (see Figure 7c); the 3-layer pleated face mask had 39% leakage (see Figure 7); Figure 7e–g shows the face masks that presented the best performance, 98% cotton, neoprene, and KN95, with 26%, 25%, and 25% leakage, respectively. Figure 7h shows the thermal spectrum of the ideal test where there was no leakage. The results were congruent with those presented in related studies [18,30].

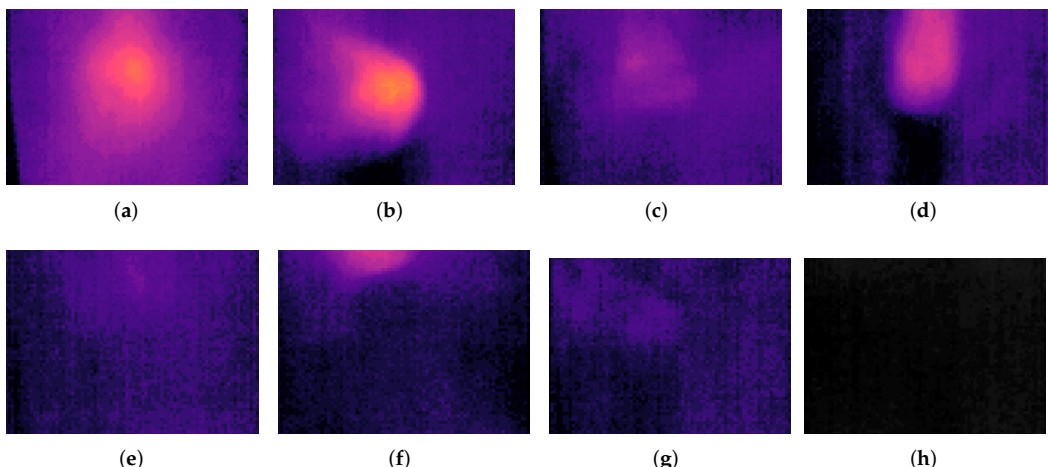

**Figure 7.** Representative thermographic spectrum images of each test captured by the imaging system. (**a**) Without a mask, 100% leakage; (**b**) 100% cotton, 69% leakage; (**c**) 100% polyester, 60% leakage; (**d**) 3-layer pleated, 39% leakage; (**e**) 98% cotton, 26% leakage; (**f**) neoprene, 25% leakage; (**g**) KN95, 25% leakage; (**h**) mask without any leak, 0% leakage.

The YOLO v4 object detection network trained with the data set previously mentioned in Section 2.5 was used to verify the proposed method's efficiency; the 311 images from the test set were analyzed. Data obtained from the mAP showed 1143 detections, and 311 unique truth values were taken into account. Figure 8 shows the loss and mAP curve during training. In the initial stage of training, the model's learning efficiency was higher, and the speed of convergence of the training curve was fast. As the training increased, the slope of the loss curve gradually decreased. Finally, when the number of training iterations reached around 5000, the model's learning efficiency reached convergence, and the loss fluctuated around 0.5. Table 4 showed each category's testing performance metrics

within the data set. Examining the confusion matrix for the performance metrics of the classification task used (Figure 9), it can be seen that one category of leakage percent was classified with a precision of 100% and one with precision more significant than 90%. The two worst-performing categories, 60%, and 26% had the most errors, including confusion between neighboring categories.

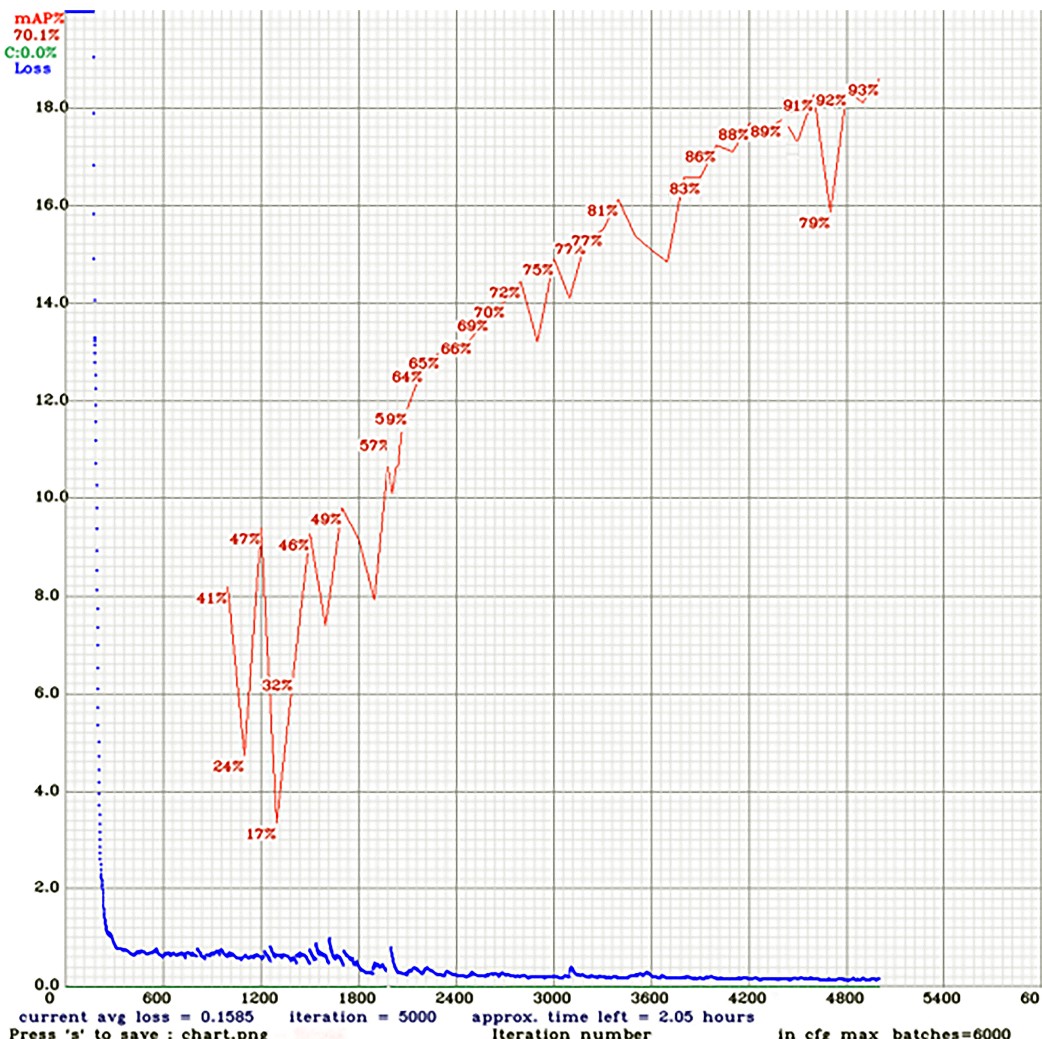

**Figure 8.** Loss and mAP curve of the YOLO v4 model.

**Table 4.** Testing performance metrics per category.

| Category | n (Truth) | n (Classified) | Accuracy | Precision | Recall | F1 Score |
|----------|-----------|----------------|----------|-----------|--------|----------|
| 100%     | 40        | 42             | 98.71%   | 0.93      | 0.97   | 0.95     |
| 69%      | 45        | 45             | 96.78%   | 0.89      | 0.89   | 0.89     |
| 60%      | 58        | 43             | 92.6%    | 0.91      | 0.67   | 0.77     |
| 39%      | 34        | 41             | 93.89%   | 0.68      | 0.82   | 0.75     |
| 26%      | 47        | 43             | 92.28%   | 0.77      | 0.70   | 0.73     |
| 25%      | 71        | 75             | 91.64%   | 0.80      | 0.85   | 0.82     |
| 0%       | 16        | 22             | 98.07%   | 0.73      | 1.0    | 0.84     |

For a minimum threshold of 0.25, a precision of 0.815, a recall of 0.842, and an F1-score of 0.821 were obtained. The proposed model correctly classified 255 images, only 56 prediction errors, and had an IoU average of 58.06%. The mean average precision (mAP@0.5) for the data set was 0.9345 or 93.45%.

**Figure 9.** Confusion matrix of testing recall.

## 4. Discussion

The global SARS-CoV-2 pandemic has led to the development and implementation of personal protective equipment (specifically face masks) to prevent the virus spread from person to person by air. This situation has allowed the creation of methods to demonstrate the effectiveness of facial masks.

This research, based on thermography and deep learning, provides certainty to discern the mask that best protects the general population and health professionals, unlike those carried out in previous research. In [18,30], the fitted filtration efficiency was measured by the Occupational Safety and Health Administration's modified ambient aerosol CNC quantitative fit testing protocol for filtering facepiece.

The results reveal that thermography represents significant advantages over other methods used to monitor the efficiency of facial masks since heat loss through the micropores is evaluated, and it represents the percentage of normal breathing that filters through the face masks, either when inhaling or exhaling. It is worth mentioning that this heat loss is proportional to the quality of the mask analyzed, that is, the higher the heat loss, the higher the leakage percentage, and the lesser the quality; the lower the percentage of leaks, the higher the quality of the face masks. As we know, previous studies have shown that the effectiveness of using masks to filter particles depends on their characteristics [17].

Thermal images are a reliable tool to evaluate heat losses through the mask; as we know, heat represents normal respiration; thus, it is an effective method that does not require meaningful equipment and can be used in future similar situations. The proposed work is a milestone for evaluating face masks for consumers and also for professional use. One of the most excellent benefits of this study is that by having a trained network and deep learning as a basis, a graphical visualization and obtaining images are fast and efficient. However, the most significant limitation is that the images are only adequate

when studied in a single individual at a time. Although thermographic vision can be used within an open population and adequate results can be obtained, it can also lead to a bias when applied en masse.

## 5. Conclusions

The thermography-based image acquisition system developed in this work proved helpful in estimating the effectiveness of the microparticle filtering of consumer-grade face masks. The exact filtration efficiency (FFE) required to prevent respiratory virus transmission was not precisely known; however, evidence from previous studies suggested that even face masks with an FFE of less than 95% (for example, surgical face masks) were effective in preventing the acquisition of epidemic coronaviruses (SARS-CoV-1, SARS-CoV-2). This article reported that heat loss through the micropores of various face mask materials substantially influenced filtration efficiency. By combining thermal imaging with deep learning tools, the consumer-grade face masks' effectiveness depending on heat loss through the manufacturing material was demonstrated. As a result, KN95 and neoprene face masks in which the micropores were virtually impenetrable were found adequate and efficient to prevent for the spread of COVID-19, unlike fabric or homemade face masks, in which the function of micropores was not adequate. These tests were designed to quantify the protection that face masks offer the wearer when exposed to other people who may be infected.

One of the proposed method's deficiencies is that if there are variations in the environmental conditions (temperature and humidity) where the images are obtained, the error in the measurement increases considerably. Improving the method accuracy is necessary to increase the test bench by a considerable amount; the added images will be constantly updated to feed the data set with the necessary information that will allow adequate network training in order to generalize the results during the processing of the measurements.

**Author Contributions:** Conceptualization, J.A.F.-M., M.P.-P. and J.L.C.-A.; data curation, Y.G.-B. and L.E.G.-R.; formal analysis, H.D.V.-D. and E.C.-C.; investigation, J.C.M.-M.; methodology, J.A.F.-M. and J.L.C.-A.; software, Y.G.-B.; supervision, M.P.-P.; validation, Y.G.-B. and E.C.-C.; visualization, J.A.G.-R.; writing—original draft preparation, J.A.F.-M., E.C.-C and J.A.G.-R. All authors have read and agreed to the published version of the manuscript.

**Funding:** This research received no external funding.

**Institutional Review Board Statement:** This study was conducted according to the guidelines of the Declaration of Helsinki, and approved by the Institutional Review Board of Tecnológico Nacional de México/I.T Tuxtla Gutiérrez (DI:523/2020).

**Informed Consent Statement:** Informed consent was obtained from all subjects involved in the study.

**Data Availability Statement:** Data are available on request.

**Acknowledgments:** I want to express my great appreciation to Tecnológico Nacional de México/I.T Tuxtla Gutiérrez and the Consejo Nacional de Ciencia y Tecnología (Conacyt) for their assistance in providing me with the resources I needed to run this work.

**Conflicts of Interest:** The authors declare no conflict of interest.

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
