# Peer review of "Towards an Approach for Filtration Efficiency Estimation of Consumer-Grade Face Masks Using Thermography"

_applsci, doi:10.3390/app12042071_

Round 1

Reviewer 1 Report

  • Minor changes in english language required
  • Row 98 – please reference the OSHA test protocol
  • Chapter 2.3 – what would be the performance levels of the tested facemasks. Since there are different performance levels for facemasks such as FFP2, FFP3, N95, N99, N100, it would be advisable to declare them in this chapter.
  • Row 123 – can you please add a sentence or two explanining how your images had a resolution of 618×450 pixels while IR camera had a resolution of 80×60 pixels
  • Equation 3 – it's clear what is FN, but please explain what is FN'
  • Please insert figures and tables just under the position in text mentioning them – especially evident for figures 3-6
  • Figure 7 – what were the level and span set when conducting IR measurements?
  • Figure 7 – what were the parameters set during the IR measurements (emissivity, reflected temperature, etc.
  • Can you please explain in detail what material is heat diffusion screen made of and what are its properties regarding infrared imaging (emissivity, transmission, etc)
  • Can you please explain how did you manage to resolve the issues of reflections and temperature change within the room which might have influenced the thermal images.
  • What was the time period between the measurements with different masks, did the diffusion screen had the chance to condition itself?

Reviewer 2 Report

The article presents the method for evaluation of quality of mask for COVID-19 prevention and is quite interesting. I like the article and the topic is very needed. In general, you have a proper academic way of referring and a good language. The aricle is clearly written and can be also used for educational purposes.

Congratulations to the authors of the work.

Comments and suggestions:
1. Please discribe your process of data augmentation in more detail.
2. Could you include a scheme of architecture of used neural network?
3. Line 177: Typo - "wer"
4. Please describe the limitations of your method.
5. Are there any ideas how to improve your method in future?

After minor revision, the article can be accepted.

Reviewer 3 Report

There are some issues to be further addressed. For example:

  1. Please give the full name of OHSA.
  2. What do i, j, m and n represent in Formula (1)?
  3. Table 1 contains the data shown in Figure 4, and it is redundant.
  4. There is a lack of explanation of calculating the average gray level of each frame used in the study.
  5. What does p (r) in formula (5) mean?
  6. Formula (7) has an expression format error. Please correct it.
  7. Why did the author use YOLO method? Please give explanation.
  8. Reference list needs to be enhanced. Some intelligent methods have potential to deal with the problem, such as

[1] Modeling an effectual multi‐section You Only Look Once for enhancing lung cancer prediction[J]. International Journal of Imaging Systems and Technology, 2021, 31(4): 2144-2157.

[2] Infrared Thermography and Computational Intelligence in Analysis of Facial Video-Records[C]//International Conference on Computational Collective Intelligence. Springer, Cham, 2021: 635-643.

[3]An Adaptive Localized Decision Variable Analysis Approach to Large-Scale Multiobjective and Many-objective Optimization, IEEE Transactions on Cybernetics, 2021,http://dx.doi.org/10.1109/TCYB.2020.3041212.

These references should be included in the paper.

Round 2

Reviewer 3 Report

My main concerns have been addressed.